# Insulin Resistance in Patients with Acne Vulgaris

**DOI:** 10.3390/biomedicines11082294

**Published:** 2023-08-18

**Authors:** Michalina Gruszczyńska, Anna Sadowska-Przytocka, Weronika Szybiak, Barbara Więckowska, Katarzyna Lacka

**Affiliations:** 1Students’ Scientific Society at Poznań University of Medical Sciences, Student’s Scientific Section of Endocrinology, Department of Endocrinology, Metabolism and Internal Medicine, Poznań University of Medical Sciences, 61-701 Poznan, Poland; michalina.gruszczynska97@gmail.com (M.G.);; 2Department of Dermatology, Poznań University of Medical Sciences, 61-701 Poznan, Poland; 3Department of Computer Sciences and Statistics, Poznań University of Medical Sciences, 61-701 Poznan, Poland; 4Department of Endocrinology, Metabolism and Internal Medicine, Poznań University of Medical Science, 61-701 Poznan, Poland

**Keywords:** acne vulgaris, insulin resistance, HOMA-IR, resistant acne

## Abstract

Acne vulgaris presents multifactorial pathogenesis, which may include insulin resistance. To investigate whether insulin resistance is a causative factor in acne vulgaris development, this cohort study and a systematic review were conducted. A cohort of 41 acne vulgaris patients and 47 healthy BMI-matched controls were recruited. Glucose and insulin fasting serum levels were obtained and the HOMA-IR was calculated; insulin resistance was diagnosed in cases with a HOMA-IR value over 2.1. The mean ± SD values for glucose fasting serum level were as follows: 94.88 ± 7.731 (mg/dL) in the study group and 79.51 ± 7.175 (mg/dL) in the controls (*p* < 0.001). The mean ± SD insulin fasting serum levels were 14.47 ± 6.394 (µIU/mL) and 11.83 ± 4.309 (µIU/mL) (*p* = 0.059), respectively. The HOMA-IR mean ± SD value calculated for the study group was 3.4 ± 1.49 and, in the control group, it was 2.34 ± 0.909 (*p* < 0.001). Out of 41 patients, 32 were diagnosed with insulin resistance (78%), and 26 of the 47 controls were diagnosed with IR (55%) (*p* = 0.026). An insulin resistance diagnosis was statistically more common among the acne patients, compared to the controls. In the articles reviewed in this paper, insulin resistance was found to be more frequent in acne vulgaris patients. Both our study and the papers analyzed in the review indicate that insulin resistance might be an independent factor in acne vulgaris development and should be considered when diagnosing and treating acne.

## 1. Introduction

Acne vulgaris is a very common skin disorder and its prevalence is still increasing. The first symptoms usually occur at the age of 11–12 years old and, eventually, up to 85% of adolescents are affected. Acne vulgaris is a growing problem in developed countries and is rare in developing countries. The reason for the lower incidence of acne lesions in patients from less affluent countries is a diet containing less milk and fewer dairy products and carbohydrates [1]. Western diets, which contain food with a high glycemic index, have a significant impact in terms of disturbances in glucose and insulin serum levels and are associated with the development of insulin resistance (IR).

In severe cases, acne may have an impact on skin condition in adulthood, especially when severe lesions do not heal properly, leading to discolorations and scars. The number of adult patients with acne vulgaris continues to increase. Moreover, both acne and its complications may reduce self-confidence and induce mental health problems [2]. It is worth highlighting that the occurrence of extensive acne lesions affects the quality of life and is a factor that influences psychosociological health. The correlation between psychosocial disorder and acne vulgaris is more significant among women than in the male population [3].

The development of acne lesions is strongly associated with metabolic and hormonal disorders. Acne vulgaris may be induced in the course of disorders that are characterized by abnormal levels, not only of androgens, estrogens, and progesterone but also of insulin and insulin-like growth factor-1 [4,5]. Acne vulgaris may be regarded as a civilization-related disease, as is IR. The skin often reflects the body’s internal health; therefore, acne formation may be associated with IR. IR is a condition characterized by the failure of insulin to provide the proper glucose transport from the bloodstream into the tissues, which results in the development of hyperglycemia and hyperinsulinemia [2].

Recently, a growing level of interest concerning a possible relationship between these diseases has been observed, although thus far, little information is available. What is already clear is that insulin serum levels have been observed to be elevated in patients with acne vulgaris, and that both acne and IR share the hormonal and signal transduction pathways, including insulin-like growth factor-1 (IGF-1) and the mammalian target of rapamycin kinase 1 (mTORC1) [6]. Furthermore, both acne and IR are observed in the course of syndromes such as HAIR-AN or hyperandrogenism (HA), IR, and acanthosis nigricans (AN) or polycystic ovary syndrome (PCOS) [6]. Several studies have revealed noticeably higher insulin serum levels and HOMA values among acne patients compared to the healthy controls; therefore, they indicate the importance of considering IR as a causative factor in acne formation [7,8,9,10,11,12]. However, as yet, no research has been conducted on the Polish population and more data are required in order to include IR evaluation as part of standard acne diagnostics and treatment options.

In this project, we aimed to investigate IR in patients with acne vulgaris and compare our results with the data from the different papers included in our systematic review.

## 2. Materials and Methods

### 2.1. Study Design and Participant Recruitment

The study was conducted in the outpatient section of the Department of Dermatology and Venerology of Poznań University of Medical Sciences in Poland.

A total of 41 patients diagnosed with acne vulgaris and 47 controls were enrolled in the study. Out of the 41 patients, 38 were female (93%) and 3 were male (7%). Their mean age was 30.63 ± 10.57 and their mean BMI was 24.57 ± 3.867. The control group comprised 47 people, of whom 24 were female (51%) and 23 were male (49%). Their mean age was 25.87 ± 3.9 and their mean BMI was 22.9 ± 2.53. Informed consent was received from each participant. The inclusion criterion for both the study and control groups were male and female patients aged 18–65. The inclusion criterion for the study group was the presence of acne vulgaris. Patients with lesions evaluated as being either moderate or severe were involved in the study group. Examples of the severity of acne lesions in the study group are presented in Figure 1.

The exclusion criteria for the study group were the presence of any hormonal disorders, including polycystic ovary syndrome, thyroid dysfunction, Cushing’s syndrome, congenital adrenal hyperplasia, and diabetes mellitus (or a family history of DM). Moreover, patients on any hormonal treatment, including hormonal contraceptive therapy or treatment with drugs that may decrease insulin sensitivity, including antipsychotic drugs, e.g., olanzapine, clozapine, risperidone, or quetiapine, and statins and chronic treatment with niacin (vitamin B3) were excluded [13]. The exclusion criterion in the control group was the presence of any chronic disease, including acne vulgaris, or any chronic treatment. The exclusion criterion in both groups was a high intake of high-glycemic-load carbohydrates and milk and dairy products.

Subjects were asked to complete a questionnaire, after which, a physical examination was performed. The severity of the patient’s acne was evaluated using the investigator’s global assessment scale. A physical examination was performed in the control group and included measurements of the subject’s height, weight, waist and hip circumference, heart rate, and blood pressure.

The blood samples from the peripheral venous blood were collected after overnight fasting. The glucose serum level was evaluated in the central hospital laboratory. Insulin serum level was evaluated using the enzymatic ELISA DRG method. The normal ranges for these parameters are as follows: glucose 70–99 and insulin 5–25 µIU/mL. The homeostasis model assessment of IR (HOMA-IR) was calculated using the following formula: fasting insulin serum level (mU/mL) × fasting glucose serum level (mmol/L)/22.5. According to the criteria established for a Polish population by Szurkowska M. et al. in 2005, values over 2.1 indicated the presence of IR [14]. The value was evaluated as the third quartile of HOMA-IR values, calculated for a healthy population with a BMI < 25 kg/m^2^.

### 2.2. Analysis

Statistical analysis was performed using IBM SPSS Statistics 29.0.0.0. For continuous data, the normal distribution was evaluated using the Shapiro–Wilk test, while variance equality was evaluated using the Fisher–Snedecor test. For continuous data with a normal distribution and equal variances, an unpaired *t*-test was performed to compare the groups; when the variances were different, the Cochran–Cox correction of the *t*-test was performed, and when both assumptions were not met, the Mann–Whitney test was performed. A comparison of the nominal data was made using the chi-square test. All numeric variables are presented as the mean ± standard deviation (SD) and nominal values as the frequency and percentage of each category. A *p*-value of <0.05 was accepted as indicating statistical significance.

### 2.3. Database Search for Systematic Review

The aim of our systematic review was to analyze and compare our results with other original papers describing IR in patients with acne. The PubMed, Google Scholar, Scopus, and Cochrane Library databases were searched independently by two authors. The search strategy included the use of controlled keywords and vocabulary, using the following terms: “insulin resistance”, “acne vulgaris”, “insulin resistance in patients with acne vulgaris”, “HOMA-IR”, and “HOMA-IR in acne vulgaris”.

The initial selection of studies was based on the abstracts and keywords. Subsequently, full-text versions were obtained. The clinical importance of the papers obtained was considered and so case reports or papers presenting a study group only were excluded, due to their low clinical importance. Moreover, Greenhalgh’s evidence hierarchy was applied. Studies published between 2012 and 2022 were considered for this review. The PRISMA list was followed during the construction of the review and the systematic literature research in this study was developed according to PRISMA guidelines.

To include articles in our systematic review, we decided to apply the following selection criteria: an original paper, using the same method of examination of insulin serum levels, and the HOMA-IR and results being presented as a mean ± SD or median with a range. To exclude any possible impact from other abnormalities on IR development, we decided to apply the following exclusion criteria: the presence of PCOS or other endocrinopathies and treatment with drugs affecting insulin sensitivity. The process for study selection is presented in Figure 2.

We highlighted 11 articles dealing with IR and acne vulgaris that determine a connection between these conditions. Those results presented as either mean ± SD or median (range) were considered separately and are shown in Table 1 and Table 2, respectively.

The most significant results, characterized by a *p*-value of <0.001, were obtained by Snight M. et al. and Nurhadi S. et al., who recorded HOMA-IR values in the acne group (3.3 ± 1.7) and (2.63 ± 0.29), and in the control group (1.5 ± 1.9) and (1.71 ± 0.26), respectively.

The study was approved by the Ethics Committee of the Poznań University of Medical Sciences in Poland.

This research project was implemented with the use of funds for science awarded by the Poznań University of Medical Sciences.

## 3. Results

### 3.1. Descriptive Analysis

Groups of 41 patients and 47 healthy controls were age- and BMI-matched (*p* > 0.05). The mean BMI in both groups was within the normal range (BMI < 25 (kg/m^2^)). There were 18 (44%) acne patients and 9 controls (19%) with a BMI calculated as ≥25 (kg/m^2^).

The mean glucose fasting serum level in the study group was 94.88 (mg/dL), whereas in the control group, it was 79.51 (mg/dL), which was statistically significant (*p* < 0.001). The mean insulin fasting serum level in the study group was 14.47 (µIU/mL) and 11.83 (µIU/mL) in the controls and it had no statistical significance (*p* = 0.059). The HOMA-IR mean value calculated in the study group was 3.4 and in the control group, it was 2.34, with a strong statistically significant difference between the two groups (*p* < 0.001). Assuming the cut-off value for IR determination to be 2.1, 32 out of the 41 patients were diagnosed with IR (78%) and, out of the 47 controls, 26 were diagnosed with IR (55%), which was statistically significant (*p* = 0.026).

Regarding the calculation formula for determining the HOMA-IR cut-off value, we would like to emphasize that in our study method, in the case of our healthy control group, when excluding people with a BMI of ≥25 kg/m^2^, the third quartile of HOMA-IR values was estimated as 2.69. However, a study with a bigger group is necessary.

Nevertheless, if the HOMA-IR cut-off value for our study was established as 2.69, then, 26 out of the 41 patients would be diagnosed with IR (63%), and 14 out of the 47 controls would be diagnosed with IR (30%), with the difference being even more significant (*p* = 0.002). The results of the calculated HOMA-IR, shown as the percentage of patients diagnosed with IR in both groups, are presented in Figure 3.

Our results demonstrated that the glucose serum level and HOMA-IR were significantly higher in the acne vulgaris group compared to the control group, which means that IR might be an independent causative factor in acne vulgaris formation. All the results are presented in Table 3.

### 3.2. Systematic Review Results

#### 3.2.1. Insulin Serum Levels

Upon analyzing the data included in Table 1 concerning the serum levels of insulin, it is remarkable that in seven out of the nine articles, the mean insulin serum levels were higher in the group of patients with acne; of these articles, four sets of study results were statistically significant, which was assessed by a *p*-value of <0.05. The most significant difference between the mean insulin serum levels in the study group (33.85 ± 1.5) and the control group (7.14 ± 2.4) was demonstrated in the study by Snight M. et al., in which the *p*-value was calculated as *p* < 0.001. One of the studies did not include information about insulin serum levels. As shown in Table 2, only one of the selected studies contains information about insulin serum levels, presented as a median with a range. In this case, there was no significant difference between acne patients and healthy controls.

#### 3.2.2. HOMA-IR

Comparing the data from Table 1 on HOMA-IR values in the study group to those of the control group, it is noticeable that in eight out of nine articles, HOMA-IR was higher in patients with acne compared to the controls. In six studies, the results were statistically significant, which was assessed by a *p*-value of <0.05. The most significant results, characterized by a *p*-value of <0.001, were obtained by Snight M. et al. and Nurhadi S. et al., who recorded HOMA-IR values in the acne group (3.3 ± 1.7) and (2.63 ± 0.29) and in the control group (1.5 ± 1.9) and (1.71 ± 0.26), respectively. Table 2 details two studies presenting data as a median with a range, only one of which showed higher HOMA-IR values being calculated for the study group, compared to the controls. The results published by Mustafa AL et al. showed a strongly significant difference between acne patients (6.02 (5.2–8.1)) and the controls (3.51 (2.2–4.3)), with a calculated *p*-value of <0.001.

## 4. Discussion

### 4.1. Acne Vulgaris

Acne vulgaris is a chronic inflammatory skin disorder that involves the folliculopilosebaceous unit of the skin. It is a relatively common condition, as can be seen on the skin of 85% of teenagers and 33% of 15–44-year-old patients [1]. It most often affects female adolescents and adult women, although there is a high prevalence among males [9]. Acne lesions, which include comedones, papules, pustules, and nodules, are most often located within the skin of the face, shoulders, chest, and back [6]. Many different factors may lead to acne formation, including abnormal sebum production (hyperseborrhea), hyperkeratosis, inflammation, or hormonal disturbances. Elevated levels of androgens, insulin, and insulin-growth factor-1(IGF-1) lead to the promotion of hyperkeratosis and hyperseborrhea, resulting in acne formation [20]. Moreover, these hormones affect each other, e.g., androgens lead to an increase in insulin and IGF-1 serum levels, whereas the androgen signal transduction is stimulated by IGF-1. In addition, IGF-1 decreases the serum level of sex hormone binding globulin (SHBG), leading to the elevation of free androgens [6].

### 4.2. Insulin Resistance

IR is a condition that is characterized by the failure of insulin to provide the proper glucose transport into the tissues, which results in the development of hyperglycemia and hyperinsulinemia [21]. IR may be triggered by many factors, including genetic factors. However, environmental factors seem to play a major role in IR development.

Insulin and IGF-1 serum levels may be elevated in the course of many endocrine disorders, including acromegaly, hyperprolactinemia, hypercortisolism, or congenital adrenal hyperplasia [21]. Obesity, chronic inflammation, hypertriglyceridemia, or the activity of antagonistic hormones, including glucagon, cortisol, and thyroxin, seem to be major causative factors in IR development [22]. Western diets, which include a high intake of high glycemic-load carbohydrates and milk, promote sudden fluctuations in glucose and insulin serum levels [16]. Moreover, milk itself contains IGF-1, promoting IR development in various different ways. Increased milk and high-glycaemic-load product intake has been detected in acne patients compared to healthy controls; moreover, no acne lesions were observed in populations following a Paleolithic style of eating [22]. An important fact is that high insulin and IGF-1 play a significant role in the development of type 2 diabetes and cardiovascular diseases [23]. All of these are diseases associated with civilization, which emphasizes the importance of IR diagnosis and treatment. IR therapy includes both lifestyle changes—a low glycemic-load diet and physical activity—and metformin therapy.

There is a wide range of methods to select from when evaluating IR. The euglycemic metabolic clamp technique was the first method, which was designed by R. A. DeFronzo, J. D. Tobin, and R. Andres in 1979 but, due to its complicated process consisting of both glucose and insulin infusions, it is currently not widely used [24]. The homeostasis model assessment scale (HOMA) is probably the one most frequently used in IR calculations. Only the fasting serum levels of both glucose and insulin are required for the formula. However, the different cut-off values calculated in various populations, starting from 1.7 for a Japanese population, up to 3.8 for a French population, make it difficult to interpret [25,26]. Similar indexes include the quantitative insulin sensitivity check index (QUICKI) or the Matsuda index, which requires three measurements of glucose and insulin serum levels to be taken during an oral glucose tolerance test (OGTT) [27,28]. An innovative idea for insulin sensitivity evaluation is to measure irisin serum levels, which have been found to present a significant negative correlation with HOMA-IR values [29].

### 4.3. Original Results and Systematic Review

Our study results demonstrated a correlation between insulin resistance and the presence of acne vulgaris. The mean HOMA-IR values were statistically higher in the acne group (3.40) compared to the control group (2.34); moreover, the incidence of IR for both cut-off values for HOMA-IR was statistically higher in the study group vs. the control group. The possible impact of age and obesity was excluded since the groups were age- and BMI-matched. We proved the influence of decreased insulin sensitivity on skin conditions.

The connection between IR and acne vulgaris formation has already been a point of interest among several authors, whose results were similar to those obtained in the current study.

A study by Monica Singh et al. showed statistically higher mean HOMA-IR values in the studied group of 80 acne patients (3.3), compared to 80 healthy controls (1.5). However, there was no correlation found between HOMA-IR and acne severity [7].

Sharma S et al. examined a cohort of 100 adult patients and 100 controls in mixed-sex groups. The mean HOMA-IR value in the acne group (2.7) was found to be statistically higher than that in the control group (1.9) [8].

Similarly, the study by Mustafa et al. proved the influence of IR in acne vulgaris formation when the median HOMA-IR value in a mixed-sex, 60-patient acne group (6.02) was statistically higher than in an identical non-acne group (3.51) [19]. An interesting discovery was a negative correlation between the serum irisin level, which is a natural obesity, diabetes, and IR protector, and HOMA-IR among acne patients; therefore, the irisin serum level was lower in acne patients compared to the controls. Moreover, the level had a negative correlation with acne severity. Physical exercise, resulting in fat reduction and weight loss, increases the serum irisin level [30].

Nagpal et al. examined 100 male patients and 100 controls and recorded statistically higher HOMA-IR mean values in the acne group (2.0) than in the non-acne controls (1.7). Any possible impact of hyperandrogenemia was excluded [9].

Moreover, statistically higher mean HOMA-IR values were observed in the studies by Nurhadi et al. [10] (an acne patient value of 2.63 vs. a non-acne patient value of 1.71), in the study by Emiroglu N et al. [11] (2.87 vs. 1.63), and in the study by Del Prete et al. [17] (1.7 vs. 1.1). In contrast, Cerman et al. did not observe any connection between IR and acne vulgaris formation [16].

Aydin et al. [15] and Cetinozman et al. [17] obtained mean HOMA-IR values from their study groups that were higher than in the controls, although there was no statistical significance.

In the study by Nazik H. Hazrat et al. in 2023, an original method of IR evaluation was used. Serum levels of C-peptide, triglyceride, and glucose were obtained; later, a TyG (triglyceride-glucose) index was calculated using the formula: {Ln (natural logarithm) (TG [mg/dL] × glucose [mg/dL]}. Values over 1.89 ng/mL for C peptide and over 4.49 for the TyG index indicated a diagnosis of IR. As in other studies, these results suggested statistically more frequent IR development in the group of acne patients, compared to the healthy controls [31].

The study by Solanki et al. examined only the study group, although it investigated the original topic of recurrent acne and hirsutism. Patients were divided into groups, based on the presence of acne, hirsutism, or both. Moreover, each of the three groups was then divided into two, consisting of either naïve patients or patients with recurrent symptoms. In every group, the percentage of IR patients was statistically higher among patients with recurrent symptoms compared to the naïve subjects. The results suggested a strong correlation between IR and recurrent acne and hirsutism [32].

Another innovative study was conducted by Kaya İFK et al., in which researchers studied the influence of IR and visceral fat on acne formation. Apart from the HOMA-IR index, the VAI (visceral adipose index) was used. For the VAI calculation, the values of waist circumference, body mass index (BMI), and TG and HDL serum levels were required. However, there was no significant difference found between the acne group and the control group, in terms of HOMA-IR and VAI [18].

Mustafa AI et al. studied the more limited topic of post-adolescent acne. They proved that IR was more frequent in the acne group than in the controls. Moreover, a strong predictor for post-adolescent acne formation could be serum fetuin-A concentration, which was statistically higher in the acne group. Furthermore, HOMA-IR and fetuin-A serum levels were statistically higher in the case of patients who developed severe forms of acne, which suggested that it may be a predictor of acne severity. Fetuin-A and HOMA-IR were found to be positively correlated; this could be because fetuin-A is a glycoprotein that acts as an antagonist to the insulin receptors, eventually leading to a decrease in muscle and hepatic insulin signaling and IR development [33].

### 4.4. Metformin Treatment

IR evaluation is important in diagnosing acne vulgaris. In such cases, causal treatment could be applied, including both lifestyle changes and pharmacotherapy. Lifestyle changes, e.g., a low glycemic-load diet, were found to reduce serum levels of IGF-1 and decrease the severity of acne lesions. Recently, growing interest in possible acne treatment modalities has been observed, and metformin was noted to be an effective therapeutic method in acne treatment in such cases. Its effectiveness has already been proved in the case of female patients with polycystic ovary syndrome (PCOS); however, the treatment could probably be no less successful in male patients and in females with no PCOS. Metformin not only increases insulin sensitivity when inhibiting mTORC1 (mammalian target of rapamycin complex 1) and reducing IGF-1 (insulin-like growth factor-1) serum levels, leading to androgen signaling suppression, but also presents anti-inflammatory properties that eventually cause a reduction in the GAGS score (the global acne grading system) and the disappearance of acne lesions [34]. Metformin’s effectiveness in acne vulgaris treatment was described in the study by Robinson et al., who used a 12-week combination therapy consisting of 2.5% benzoyl peroxide and tetracycline at doses of 250 mg in both acne groups, with an extra metformin dose of 850 mg in the study group. Patients treated with metformin obtained better treatment results compared to the no-metformin group, which proves metformin’s effectiveness in acne vulgaris therapy [35].

## 5. Conclusions

In our study, we found a correlation between IR and the presence of acne vulgaris. People with decreased insulin sensitivity could be more prone to developing acne lesions. In view of the fact that IR might be an independent factor in acne vulgaris formation, especially in forms of acne that are resistant to classical treatment methods, we highlight the importance of finding a possible relationship between these disorders when diagnosing acne and, in such cases, of focusing more on causal rather than symptomatic treatment. Both lifestyle changes and pharmacological treatment, including, e.g., metformin therapy, could be effective in such cases. However, more studies are necessary to prove its effectiveness in the treatment of acne vulgaris lesions.

## Figures and Tables

**Figure 1 biomedicines-11-02294-f001:**
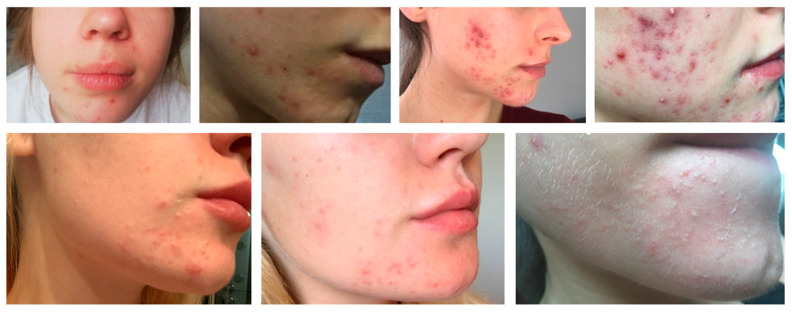
Examples of severity of acne lesions in patients included in the study group.

**Figure 2 biomedicines-11-02294-f002:**
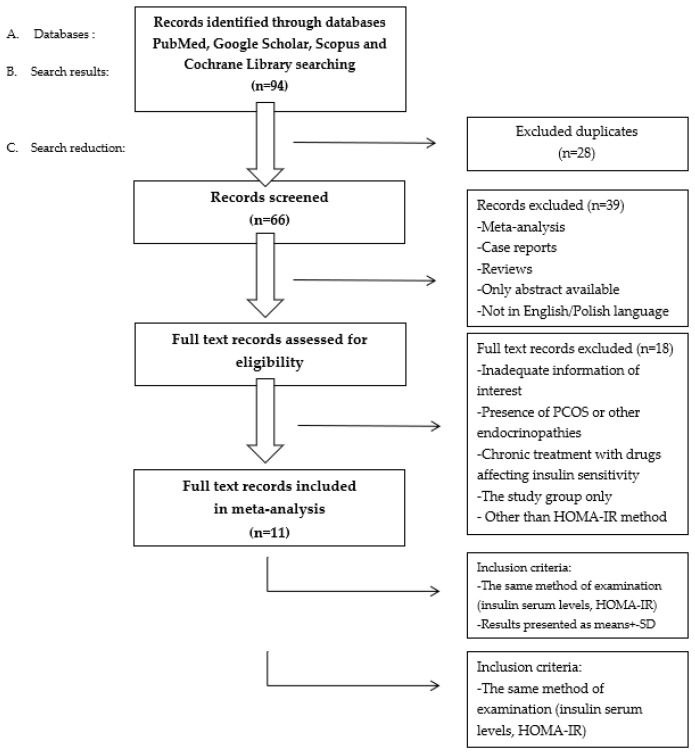
The database search flowchart. PCOS—polycystic ovary syndrome, HOMA-IR—homeostasis model assessment of IR, SD—standard deviation.

**Figure 3 biomedicines-11-02294-f003:**
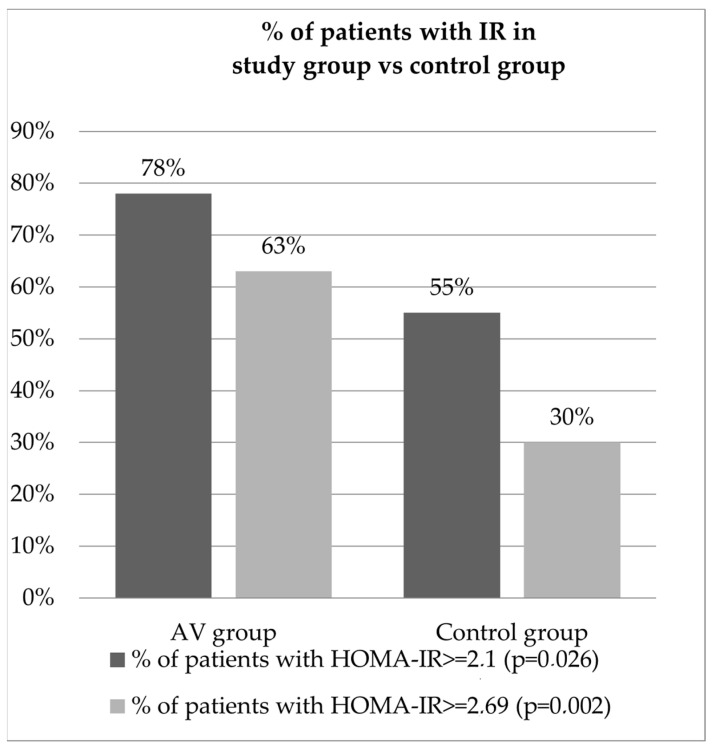
The percentage of patients with IR in the acne vulgaris group, compared to the control group.

**Table 1 biomedicines-11-02294-t001:** Study results, presented as mean ± standard deviation.

Author, Year of Publication	Group	Number		Insulin (miU/L)		HOMA-IR	
Singh M et al., 2022 [7]		F	M	Mean ± SD	*p*-Value	Mean ± SD	*p*-Value
	Acne	22	58	33.85 ± 1.5	<0.001 *	3.3 ± 1.7	<0.001 *
	Control	33	47	7.14 ± 2.4		1.5 ± 1.9	
Sharma S et al., 2019 [8]	Acne	56	44	12.5 ± 7.7	0.000 *	2.7 ± 1.8	0.000 *
	Control	57	43	8.7 ± 4.9		1.9 ± 1.2	
Nagpal et al., 2018 [9]	Acne	0	100	9.2 ± 8.5	0.22	2.0 ± 1.8	0.04 *
	Control	0	100	7.8 ± 6.8		1.7 ± 2.3	
Nurhadi S., 2018 [10]	Acne	28	10	-	-	2.63 ± 0.29	<0.001 *
	Control	28	10	-		1.71 ± 0.26	
Aydin et al., 2017 [15]	Acne	18	0	10.3 ± 3.3	0.127	2.2 ± 0.7	0.182
	Control	18	0	8.5 ± 3.6		1.8 ± 0.8	
Cerman et al., 2016 [16]	Acne	28	22	9.72 ± 6.26	0.384	2.03 ± 1.49	0.37
	Control	22	14	9.88 ± 4.3		2.05 ± 0.93	
Emiroglu N et al., 2014 [11]	Acne	144	99	14.01 ± 11.99	0.001 *	2.87 ± 2.57	0.001 *
	Control	111	45	9.12 ± 3.55		1.63 ± 0.66	
Cetinozman et al., 2013 [17]	Acne	26	0	10.8 ± 4.6	0.1	2.1 ± 1.0	0.23
	Control	21	0	8.9 ± 2.8		1.8 ± 0.6	
Del Prete et al., 2012 [12]	Acne	0	22	10.6 ± 8.4	0.01 *	1.7 ± 0.8	0.016 *
	Control	0	22	5.5 ± 1.4		1.1 ± 0.3	

* Significant *p*-value.

**Table 2 biomedicines-11-02294-t002:** Study results, presented as median values (range).

Author, Year of Publication	Group	Number		Insulin (miU/L)		HOMA-IR	
Kaya IFK et al., 2022 [18]		F	M	Median (range)	*p*-Value	Median (range)	*p*-Value
	Acne	85	7	9.16 (4.98–17.6)	0.877	1.749 (1.005–3.573)	0.704
	Control	80	12	9.33 (5.27–14.5)		1.78 (1.040–3.038)	
Mustafa AL et al., 2018 [19]	Acne	30	30	-	-	6.02 (5.2–8.1)	<0.001 *
	Control	26	34	-		3.51 (2.2–4.3)	

* Significant *p*-value.

**Table 3 biomedicines-11-02294-t003:** Results of the current study, presented as the mean ± standard deviation.

Parameter	AV Group (*N* = 41)	Control Group (*N* = 47)	*p*-Value
Age (y) ± SD	30.63 ± 10.571	26.26 ± 3.812	0.384
SEX, N, (%)			<0.001
female	38 (93%)	24 (51%)	
male	3 (7%)	23 (49%)	
BMI ^1^ (kg/m^2^) mean ± SD	24.57 ± 3.867	22.9 ± 2.53	0.051
Glucose serum level (mg/dL), mean ± SD	94.88 ± 7.731	79.51 ± 7.175	<0.001
Glucose serum level (mmol/L), mean ± SD	5.27 ± 0.429	4.42 ± 0.399	<0.001
Insulin serum level, mean ± SD	14.47 ± 6.394	11.83 ± 4.309	0.059
HOMA-IR ^2^, mean ± SD	3.40 ± 1.49	2.34 ± 0.909	<0.001

^1^ BMI—body mass index. ^2^ HOMA-IR—homeostasis model assessment of IR.

## Data Availability

The data obtained and analyzed during the current study are available from the corresponding author upon request.

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
