# Peer review of "Insulin Resistance in Patients with Acne Vulgaris"

_biomedicines, 2023, doi:10.3390/biomedicines11082294_

Round 1
Reviewer 1 Report
Well-written study that identifies an increased diagnosis of insulin resistance in patients with acne than in healthy control patients, along with a systematic review of similar studies in the literature.
Author Response
Thank you for your review.Reviewer 2 Report
There appears to be a link between acne and insulin resistance. The relationship between acne and insulin resistance can be understood through the following mechanisms:
1. Inflammation: Insulin resistance can trigger chronic low-grade inflammation in the body. Inflammation is a key factor in the development of acne, as it promotes the production of certain substances that contribute to the formation of acne lesions.
2. Androgens: Insulin resistance is often associated with elevated levels of androgens (male hormones) in both males and females. Androgens stimulate the sebaceous glands in the skin to produce more oil, known as sebum. Excess sebum production can clog pores, leading to the development of acne.
3. Insulin-like Growth Factor-1 (IGF-1): Insulin resistance is linked to increased levels of IGF-1 in the body. IGF-1 is a hormone that plays a role in cell growth and proliferation. Elevated levels of IGF-1 can stimulate the overgrowth of skin cells within hair follicles, leading to the formation of acne.
4. Glycation: In insulin resistance, higher levels of blood sugar can lead to a process called glycation. This occurs when sugar molecules attach to proteins, forming advanced glycation end products (AGEs). AGEs contribute to skin aging and inflammation, potentially exacerbating acne.
5. Dietary Factors: Diets high in refined carbohydrates and sugars can lead to spikes in blood sugar levels, increasing the risk of insulin resistance. Such diets are also thought to influence acne development, likely due to their impact on insulin levels and subsequent effects on hormones and inflammation.
It's important to note that while there is evidence suggesting a connection between acne and insulin resistance, the relationship may vary among individuals. While the Homeostasis Model Assessment of Insulin Resistance (HOMA-IR) used in your research, is a widely used and convenient method to estimate insulin resistance, it does have some limitations that should be considered when interpreting its results:
1. Estimation, not direct measurement: HOMA-IR is an indirect method of estimating insulin resistance based on fasting glucose and fasting insulin levels. It does not directly measure insulin resistance in tissues, and the results may not reflect the actual insulin sensitivity of specific organs or tissues.
2. Limited to fasting conditions: HOMA-IR is valid only under fasting conditions when insulin and glucose levels are relatively stable. It may not accurately represent insulin resistance in post-meal situations when the dynamics of insulin and glucose are different.
3. Population-specific cut-off values: The cut-off values for defining insulin resistance using HOMA-IR may vary depending on the population being studied. Different populations and ethnic groups may have different baseline insulin and glucose levels, leading to variations in the interpretation of HOMA-IR results.
4. Sensitivity to insulin secretion: HOMA-IR is influenced not only by insulin resistance but also by insulin secretion. In conditions where insulin secretion is impaired (e.g., type 1 diabetes or advanced type 2 diabetes with beta-cell dysfunction), HOMA-IR may not accurately reflect insulin resistance.
5. Not suitable for all individuals: HOMA-IR may not be appropriate for certain individuals, such as pregnant women or those with severe liver or kidney disease. These conditions can significantly affect insulin and glucose metabolism, potentially leading to inaccurate results.
6. Dynamic changes may not be captured: HOMA-IR provides a snapshot of insulin resistance at a specific point in time. It may not capture dynamic changes in insulin sensitivity that can occur due to factors like lifestyle modifications or medical interventions.
7. Lack of consensus on threshold values: There is no universally agreed-upon threshold value to define insulin resistance using HOMA-IR. Different studies and healthcare providers may use different cut-off points, leading to some inconsistencies in the interpretation of results.
8. Does not account for other factors: Insulin resistance is a complex metabolic condition influenced by various factors, including genetics, diet, physical activity, and body composition. HOMA-IR does not consider all of these factors in its estimation.
Despite these limitations, HOMA-IR remains a valuable tool in clinical and research settings for assessing insulin resistance, especially when more invasive or elaborate tests like the hyperinsulinemic-euglycemic clamp are not feasible. It can provide useful insights into insulin sensitivity trends in large populations and help identify individuals at risk of metabolic disorders. However, its results should be interpreted cautiously, and it is always best to complement the findings with a comprehensive evaluation by a healthcare professional.
More studies are necessary to prove the effectiveness of metformin therapy or dietary changes in acne but this study is one step in the right direction.
Author Response
Thank you for your review.Reviewer 3 Report
This is an interesting study related to the relationships between insulin resistance and acne vulgaris pathology, but some important issues need to be addressed.
1. Section Introduction, sentence “What is already clear is 58 that insulin serum levels were observed to be elevated in both disorders, and both acne 59 and IR share the signal transduction pathways, including insulin-like growth factor – 1 60 (IGF-1) and mammalian target of rapamycin kinase 1 (mTORC1)[6].” This sentence should be fixed; IGF-1 is a hormone, not signaling molecule.
2. Section Introduction, References are missing in several sentences; e.g. “Several studies have revealed noticeably higher insulin serum levels and HOMA values among acne patients compared to healthy controls, and therefore point to importance of considering IR to be a causative factor in acne formation”.
3. Section Materials and Methods, subsection 2.1. Study design and participant recruitment. You should specified criteria for acne vulgaris, as inclusion criteria of the study group, whish is the minimum for diagnosis, etc. Maybe to show a representative photo.
4. Section Materials and Methods, subsection 2.1. Study design and participant recruitment “The inclusion criteria for both groups were male and female patients aged 18-65 suffering from acne vulgaris”. It is not clear what are inclusion criteria for the control group? It should be emphasized.
5. Section Materials and Methods, subsection 2.3. Database search for systematic review. You should specified time range of included studies in this study, instead “Studies published in the past 11 years were considered for this review”.
6. Section Results. The results of this study are poor and do not show anything new in this field. It would be good to add some more results, e.g. correlations of your results.
7. Section Results. The results should not be repeated; the text and table 3 show the same results.
8. Section Discussion, subsection 4.1. Acne vulgaris. There is only one reference, besides numerous literature data. Thus, reference should be added.
9. Section Discussion, subsection 4.2. Insulin resistance. The reference should be added.
10. Discussion of results presented in this study is missing.
11. Section 5. Conclusions, “In our study, we found a positive correlation between IR and acne vulgaris development”. Results related to a positive correlation between IR and acne vulgaris are missing. In addition, this study did not investigated the development of acne vulgaris, only the presence of acne vulgaris.
12. The text of the manuscript should be improved and typos eliminated.
- The text of the manuscript should be improved and typos eliminated.
Author Response
Thank You very much for the reviews. In our reviewed version of the manuscript we made corrections according to the comments.
Below, we describe our corrections point by point:
- Section Introduction, sentence “What is already clear is 58 that insulin serum levels were observed to be elevated in both disorders, and both acne 59 and IR share the signal transduction pathways, including insulin-like growth factor – 1 60 (IGF-1) and mammalian target of rapamycin kinase 1 (mTORC1)[6].” This sentence should be fixed; IGF-1 is a hormone, not signaling molecule.
->We changed this sentence to „…share the hormonal and signal transduction pathways, including insulin-like growth factor – 1 (IGF-1) and mammalian target of rapamycin kinase 1 (mTORC1)…”
- Section Introduction, References are missing in several sentences; e.g. “Several studies have revealed noticeably higher insulin serum levels and HOMA values among acne patients compared to healthy controls, and therefore point to importance of considering IR to be a causative factor in acne formation”.
-> We added the references.
- Section Materials and Methods, subsection 2.1. Study design and participant recruitment. You should specified criteria for acne vulgaris, as inclusion criteria of the study group, whish is the minimum for diagnosis, etc. Maybe to show a representative photo.
-> We corrected the criteria for the study group, moreover the pictures presenting patients are attached.
- Section Materials and Methods, subsection 2.1. Study design and participant recruitment “The inclusion criteria for both groups were male and female patients aged 18-65 suffering from acne vulgaris”.
-> We corrected the criteria for both study and control group
- Section Materials and Methods, subsection 2.3. Database search for systematic review. You should specified time range of included studies in this study, instead “Studies published in the past 11 years were considered for this review”.
-> We corrected the sentence to “Studies published between 2012 and 2022 were considered for this review.”
- Section Results. The results of this study are poor and do not show anything new in this field. It would be good to add some more results, e.g. correlations of your results.
-> This is the first original paper basing on our collected data. This manuscript summarizes the first stage of our work.
- Section Results. The results should not be repeated; the text and table 3 show the same results.
->Our aim was to summarise the data from the text in the table to make it more clear for the readers.
- Section Discussion, subsection 4.1. Acne vulgaris. There is only one reference, besides numerous literature data. Thus, reference should be added.
-> We added the references
- Section Discussion, subsection 4.2. Insulin resistance. The reference should be added.
-> We added the references
- Discussion of results presented in this study is missing.
-> Discussion is added.
- Section 5. Conclusions, “In our study, we found a positive correlation between IR and acne vulgaris development”. Results related to a positive correlation between IR and acne vulgaris are missing. In addition, this study did not investigated the development of acne vulgaris, only the presence of acne vulgaris.
-> This is corrected.
We hope that our reviewed version of the manuscript is correct, however please do not hesitate to contact us if there are more corrections necessary.
Yours faithfully,
Michalina Gruszczynska

Round 2
Reviewer 3 Report
The authors mostly answered the issue addressed by this reviewer. I have no additional comment.